# Prediction and Generalisation Over Directed Actions by Grid Cells

**Changmin Yu[1, 2,*] Timothy E.J. Behrens[3, 4], Neil Burgess[1, 4,*]**
[1]Institute of Cognitive Neuroscience, UCL, London, UK
[2]Centre for Artificial Intelligence, UCL, London, UK
[3]Wellcome Centre for Integrative Neuroimaging, University of Oxford, Oxford, UK
[4]Sainsbury Wellcome Centre, UCL, London, UK

## Abstract

Knowing how the effects of directed actions generalise to new situations (e.g. moving North, South, East and West, or turning left, right, etc.) is key to rapid generalisation across new situations. Markovian tasks can be characterised by a state space and a transition matrix and recent work has proposed that neural grid codes provide an efficient representation of the state space, as eigenvectors of a transition matrix reflecting diffusion across states, that allows efficient prediction of future state distributions. Here we extend the eigenbasis prediction model, utilising tools from Fourier analysis, to prediction over arbitrary translation-invariant directed transition structures (i.e. displacement and diffusion), showing that a single set of eigenvectors can support predictions over arbitrary directed actions via action-specific eigenvalues. We show how to define a "sense of direction" to combine actions to reach a target state (ignoring task-specific deviations from translation-invariance), and demonstrate that adding the Fourier representations to a deep Q network aids policy learning in continuous control tasks. We show the equivalence between the generalised prediction framework and traditional models of grid cell firing driven by self-motion to perform path integration, either using oscillatory interference (via Fourier components as velocity-controlled oscillators) or continuous attractor networks (via analysis of the update dynamics). We thus provide a unifying framework for the role of the grid system in predictive planning, sense of direction and path integration: supporting generalisable inference over directed actions across different tasks.

## 1 Introduction

A "cognitive map" encodes relations between objects and supports flexible planning (Tolman [40]), with hippocampal place cells and entorhinal cortical grid cells thought to instantiate such a map (O'Keefe and Dostrovsky [32]; Hafting et al. [20]). Each place cell fires when the animal is near a specific location, whereas each grid cell fires periodically when the animal enters a number of locations arranged in a triangular grid across the environment. Together, this system could support representation and flexible planning in state spaces where common transition structure is preserved across states and tasks, affording generalisation and inference, e.g., in spatial navigation where Euclidean transition rules are ubiquitous (Whittington et al. [43]).

Recent work suggests that place cell firing provides a local representation of state occupancy, while grid cells comprise an eigenbasis of place cell firing covariance (Dordek et al. [15]; Stachenfeld et al. [38]; Sorscher et al. [37]; Kropff and Treves [26]). Accordingly, grid cell firing patterns could be learned as eigenvectors of a symmetric (diffusive) transition matrix over state space, providing a basis set enabling prediction of occupancy distributions over future states. This "intuitive planning" operates by replacing multiplication of state representations by the transition matrix with multiplication of each basis vector by the corresponding eigenvalue (Baram et al. [2]; Corneil and Gerstner [13]). Thus a distribution over state space represented as a weighted sum of eigenvectors can be updated by re-weighting each eigenvector by its eigenvalue to predict future state occupancy.

*Please send any enquiries to: changmin.yu.19@ucl.ac.uk and n.burgess@ucl.ac.uk

Fast prediction and inference of the common effects of actions across different environments is important for survival. Intuitive planning, in its original form, supports such ability under a single transition structure, most often corresponding to symmetrical diffusion (Baram et al. [2]). Here we show that a single (Fourier) eigenbasis allows representation and prediction under the many different directed transition structures corresponding to different "translation invariant" actions (whose effects are the same across states, such as moving North or South or left or right in an open environment), with predictions under different actions achieved by action-specific eigenvalues. We define a "sense of direction" quantity, i.e., the optimal combinations of directed actions that most likely lead to the goal, based on the underlying translation-invariant transition structure (e.g., ignoring local obstacles). We then show how this method could be adapted to support planning in tasks that violate translation invariance (e.g. with local obstacles), and show how adding these Fourier representations to a deep RL network improves performance in a continuous control task.

We propose that the medial entorhinal grid cells support this planning function, as linear combinations of Fourier eigenvectors and therefore eigenvectors themselves, and show how traditional models of grid cells performing path integration are consistent with prediction under directed actions. Hence we demonstrate that the proposed spectral model acts as a unifying theoretical framework for understanding grid cell firing.

## 2 "INTUITIVE PLANNING" WITH A SINGLE TRANSITION STRUCTURE

Intuitive planning represents the occupancy distribution over the state space as a weighted sum of the eigenvectors of a single transition matrix (usually corresponding to symmetric diffusion), so that the effect of one step of the transition dynamics on the distribution can be predicted by reweighting each of the eigenvectors by the corresponding eigenvalue. And this generalises to calculating the cumulative effect of discounted future transitions (Baram et al. [2]).

Specifically, consider a transition matrix, $T \in \mathbb{R}^{N \times N}$, $T_{ss'} = \mathbb{P}(s_{t+1} = s'|s_t = s)$ where $s_t$ encodes the state at time $t$ and $N$ is the number of states. Then, $T^n$ is the $n$-step transition matrix, and has the same set of eigenvectors as $T$. Specifically, the eigendecomposition of $T$ and $T^n$ are:

$$T = Q\Lambda Q^{-1}, \qquad T^n = Q\Lambda^n Q^{-1} \tag{1}$$

where each column of the matrix $Q$ is an eigenvector of $T$ and $\Lambda = diag(\sigma_P(T))$, where $\sigma_P(T)$ is the set of eigenvalues of $T$. Similarly, any polynomial in $T$, $p(T)$, shares the same set of eigenvectors as $T$ and the set of eigenvalues $\sigma_P(p(T)) = p(\sigma_P(T))$. Hence:

$$\sum_{k=0}^{\infty}(\gamma T)^k = (I - \gamma T)^{-1} = Q\mathrm{diag}(\mathbf{w})Q^{-1}, \quad \text{where } \mathbf{w} = \left\{\frac{1}{1-\gamma\lambda}, \text{ for } \lambda \in \sigma_P(T)\right\} \tag{2}$$

The resolvent form (Eq. 2) is an infinite discounted summation of transitions, which under a policy and transition structure corresponding to diffusion, is equivalent to the successor representation (SR, Fig. 1E) with discounting factor $\gamma$ (Dayan [14]; Stachenfeld et al. [38]). See Mahadevan and Maggioni [29] for a related spectral approach using Fourier decomposition of $T$ for estimating the value function. The SR has been shown to be useful for navigation via gradient ascent of the future probability of occupying the target state, and has a linear relationship with the true underlying Euclidean distances in spatial tasks (hence "intuitive planning", see Fig. 1 and Fig. 2D-E).

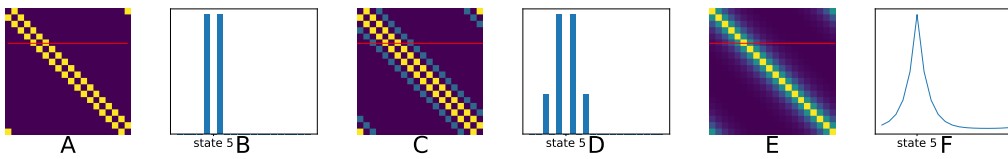

Figure 1: **Demonstration of intuitive planning on a diffusive transition task on a 1D ring track**. **A**: Example transition matrix; **B**: $\mathbb{P}(s_{t+1} = s'|s_t = 5)$; **C, D**: same are shown for $T^3$, showing predicted distribution over the next three time steps; **E** the resolvent form/SR (Eq. 2) computed from the eigenbasis of the transition matrix; **F**: SR values for state 5, which can used for navigation.

The eigenvectors of the diffusion transition matrix generally show grid-like patterns, suggesting a close relationship to grid cells. However, intuitive planning is restricted to predictions over a

single transition structure, hence cannot flexibly adjust its predictions corresponding to the effects of arbitrary directed actions (i.e., variable asymmetric transition structure), hence cannot support the presumed role of grid cells in path integration.Moreover, predictions over different directed actions would require different eigendecompositions, hence incurring high computational costs that undermines its biological plausibility. In Section 3 we unify the prediction and path integration approaches by exploiting translation invariant symmetries to generalise across actions, using a single common eigenbasis and cheaply calculated updates via action-dependent eigenvalues.

## 3 FLEXIBLE PLANNING WITH DIRECTED TRANSITIONS

Updating state representations to predict the consequences of arbitrary directed actions is an important ability of mobile animals, known as path integration and thought to depend on grid cells (McNaughton et al. [30]). To generalise the intuitive planning scheme to simultaneously incorporate arbitrary directed transition structures, we consider the transition dynamics corresponding to translation (drift) and Gaussian diffusion with arbitrary variance (including $0$, equivalent to plain translation). Our assumption that the transition structure is translation invariant (implying periodic boundary conditions), leads to circulant transition matrices.

Consider a 2D rectangular environment with length $L$ and width $W$ where each state is a node of the unit square grid, then the transition matrix can be represented by $\mathbf{T} \in \mathbb{R}^{LW \times LW}$, with each row the vectorisation ($\mathbf{vec}(\cdot)$) of the matrix of transition probabilities starting from the specified location, i.e., $\mathbf{T}[j,:] = \mathbf{vec}[\mathbb{P}(s_{t+1}|s_t = j)]$, where $\mathbf{T}$ is constructed by considering the 2D state space as a 1D vector and concatenating the rows ($j = xL + y$ for $(x, y) \in [0, W - 1] \times [0, L - 1]$), see Fig. 2A.

The transition matrix is circulant due to the translation invariance of the transition structure (see Appendix Prop. A.1), and takes the following form:

$$\mathbf{T} = \begin{bmatrix} T_0 & T_{LW-1} & \cdots & T_2 & T_1 \\ T_1 & T_0 & T_{LW-1} & \cdots & T_2 \\ \vdots & T_1 & T_0 & \ddots & \vdots \\ & & & \ddots & \\ T_{LW-2} & \cdots & \ddots & \ddots & T_{LW-1} \\ T_{LW-1} & T_{LW-2} & \cdots & T_1 & T_0 \end{bmatrix} \tag{3}$$

The normalised eigenvectors of the circulant matrix $\mathbf{T} \in \mathbb{R}^{N \times N}$ ($N = LW$) are the vectors of powers of $N$th roots of unity (the Fourier modes):

$$\mathbf{q}_k = \frac{1}{\sqrt{N}} \left[ 1, \omega_k, \omega_k^2, \cdots, \omega_k^{N-1} \right]^T \tag{4}$$

where $\omega_k = \exp(\frac{2\pi i}{N} k)$, for $k = 0, \ldots, N - 1$, and $i = \sqrt{-1}$. Hence the matrix of eigenvectors (as the columns), $F = (\mathbf{q}_0, \mathbf{q}_1, \ldots, \mathbf{q}_{N-1})$, is just the (inverse) discrete Fourier transform matrix (Bracewell [4]), where $F_{kj} = \omega_j^k$ for $0 \le k, j \le N - 1$. The Fourier modes projected back onto the $L \times W$ 2D spatial domain are plane waves, as shown in Fig. 2G, with wavevector determined by the value of $k$ that specifies the direction and spatial frequency of each plane wave (see Appendix B). We can immediately compute the corresponding eigenvalues for the eigenvectors in Eq. 4 (equivalent to taking the discrete Fourier transform (DFT) of the first row (or column) of $T$, see Bracewell [4]):

$$\lambda_m = \sum_{j=0}^{N-1} T_j \omega_j^m, \quad \text{for } m = 0, \ldots, N - 1 \tag{5}$$

where $\{T_0, \ldots, T_{N-1}\}$ are the $N$ unique elements that fully specifies the circulant matrix $T$ (Eq. 3).

We can then utilise tools from Fourier analysis for efficient updating of the eigenvalues whilst leaving the universal eigenbasis unaffected. For a transition matrix $\mathbf{T^v}$ corresponding to an arbitrary action (translation velocity) $\mathbf{v} = (v_x, v_y)$, each row of $\mathbf{T^v}$ is again a circulant, but shifted version of the corresponding row vector of the symmetric transition matrix corresponding to zero drift velocity, $\mathbf{T}^0$. Specifically, the first rows of the two matrices are related as follows:

$$\mathbf{T^v}(k) = \mathbf{T}^0(k + v_x L + v_y), \quad \text{for } k = 0, \ldots, N - 1 \tag{6}$$

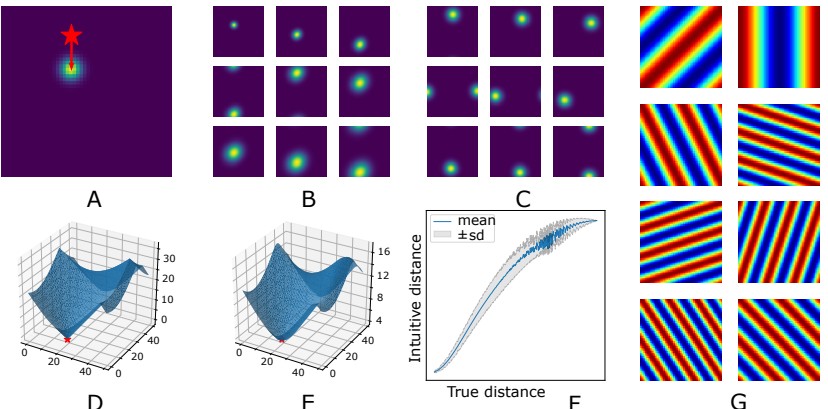

Figure 2: **Demonstration of our method in a 2D $50 \times 50$ environment with periodic boundary conditions**. **A**: Transition matrix $\mathbb{P}(x_{t+1} = (i', j')|x_t = (i, j))$ starting from a randomly chosen state $((8, 10)$; red star) with drift velocity 10 units southward and Gaussian diffusion (red arrow); **B**: Usage of the eigenbasis (Eq. 4) and our analysis (Eq. 7) for predicting the distribution over future states given the transition structure given in A, showing successive changes to state occupancy; **C**: Application of our model to translation-only transition structures (drift velocity $v = (3, 3)$); **(D-E)**: Ground-truth shortest distance to state $(8, 20)$ (red star; D) as a function of the starting location, and distance estimated under the intuitive planning framework (E) using plain diffusion; **F**: Estimated distance measure (E) v.s. the corresponding ground-truth distance (D) over all pairs of states; **G**: Examples of 2D Fourier modes (real parts shown, see Appendix Fig. 7 for the top 100 eigenvectors).

Given the eigenvalues for $\mathbf{T}^0$, $\Lambda^0 = \left[\lambda_0^0, \lambda_1^0, \ldots, \lambda_{N-1}^0\right] \in \mathbb{C}^N$ (via the DFT of the first row of $\mathbf{T}^0$, Eq. 5), we can immediately derive the eigenvalues of $\mathbf{T}^{\mathbf{v}}$, $\Lambda^{\mathbf{v}}$, via a one-step update based on the Fourier shift theorem (Bracewell [4]) without recomputing the eigendecomposition:

$$\Lambda^{\mathbf{v}}[k] = \exp\left(\frac{2\pi i}{N}(v_x L + v_y)k\right)\Lambda^0[k], \quad \text{for } k = 0, \ldots, N-1, \quad \text{for arbitrary } \mathbf{v},$$

$$\text{i.e., } \Lambda^{\mathbf{v}} = \Phi_{\delta\mathbf{v}}\Lambda^0, \quad \Phi_{\delta(\mathbf{v})} = \left[1, \omega_{\delta(\mathbf{v})}, \omega_{\delta(\mathbf{v})}^2, \ldots, \omega_{\delta(\mathbf{v})}^{N-1}\right], \quad \text{where } \delta(\mathbf{v}) = v_x L + v_y \tag{7}$$

This allows path integration by reweighting the common set of eigenvectors at each timestep by the updated eigenvalues corresponding to the current drift velocity (Eq. 7). Note that additionally, $T^0$ can include diffusion, thus reweighting by the eigenvalues of the diffusive transition matrix also allows tracking of increasing uncertainty.

Utilising the fixed eigenbasis (Eq. 4) and the respective eigenvalues (Eq. 7) for arbitrary transition structures, we can make efficient prediction for the distribution of future state occupancy with respect to arbitrary action (see Figs. 2B-C). Adding translation to the translation-invariant transition matrix does not change the set of eigenvectors - allowing one set of eigenvectors (Fourier modes) to support prediction for actions in all directions (or plain diffusion), hence prediction of effects of directed actions can be efficiently generalised across environments.

**Sense of Direction.** We define a "sense of direction", $\theta^*$, as the angle of the transitions (or the linear combinations of the available actions in a non-spatial setting) that maximise the future probability of reaching the target state given an initial state, which is modelled by the SR matrix.

$$\theta^* = \arg\max_{\theta} \sum_j \frac{\exp[2\pi i(\mathbf{x}_G - \mathbf{x}_0) \cdot \mathbf{k}_j]}{1 - \gamma D_j \exp[2\pi i \mathbf{v}_\theta \cdot \mathbf{k}_j]} \tag{8}$$

where $\gamma$ is the discounting factor, $D_j, j = 1, \cdots, LW$ are the eigenvalues for the symmetric diffusion transition matrix, $\mathbf{k}_j, j = 1, \ldots, LW$ are the wavevectors for the $j$-th Fourier components, $\mathbf{x}_0, \mathbf{x}_G$ are the coordinates of the start and goal states, and $\mathbf{v}_\theta = (v\cos(\theta), v\sin(\theta))$ represents the velocity (with speed $v$ and head direction $\theta$). We see that the "sense of direction" supports generalisation of predictions of effects of actions across all environments with the same translation-invariant transition

structure, i.e., such predicted effects ignore any local deviations from translation invariance. See Appendix B for the derivation of Eq. 8. Note that here we assume that the goal state $\mathbf{s}_G$ is known a priori, e.g., we consider a problem where the animal is navigating towards a previously visited location. The derived analytical expression for the sense of direction can be retrieved via a lookup table when the state space is small and discrete, whereas in large or continuous state spaces, it can be computed either via optimisation algorithms, or modelled by a non-linear function approximator that represents Eq. 8. See Bush et al. [11] for neural network approaches to finding goal directions from grid representations.

We thus propose that a computational role for the neural grid codes: generating a "sense of direction" (capturing the transition structure of the state space, ignoring the obstacles and boundaries) that reflects a global sense of orientation which allows generalisation to completely new environments.

**Flexible Planning & Application Beyond Translation-Invariant Structures.** The proposed model can be applied to flexible planning under arbitrary drift velocity as demonstrated in Fig. 3 (A-E). An agent is trying to navigate towards a goal state in a windy grid world. The navigation is performed by following the ascending "gradient" of the SR for occupancy of the target state (the resolvent metric, Eq. 2). The SR computed from the transition matrix including the effects of diffusion and wind (Fig. 3 B) based on our analysis (eq. 7) leads straight to the target (Fig. 3 C).

Given the analytical expression of the SR (Eq. 2), we could efficiently adjust the SR matrix to accommodate local changes in the state space, e.g., insertion of a barrier, using the Woodbury inversion formula to update the parts of the SR matrix affected by the local obstacles (see Appendix A.3 for derivations [34]); and again in this case, the agent correctly adjusts for the wind as well as taking the shortest path around the inserted wall (Fig. 3 D-E).

We note, however, that the proposed model is also able to solve tasks without periodic boundary conditions, by considering the original task state space, $S_0$, being embedded into a larger, periodically bounded pseudo state space $S_p$, at least twice as large in each dimension as $S_0$ (Fig. 3 F). We again follow the previous procedures, utilising the Fourier modes, this time computed on $S_p$, to perform predictions in $S_0$ (Fig. 3 F-G), and the performance is unaffected. Note that under such formulation, the underlying transition structures can be applied to environments with both periodic and non-periodic boundary conditions - allowing sense of direction planning in either case.

**Path Integration.** We can also use our model for path integration (see also Section 4) in $S_0$, by taking velocity inputs (given any path in the grid world) to update the state occupancy distribution (Eq. 7). The path integration performance is strongly correlated with the degree of uncertainty (i.e., the diffusion strength caused by self-motion noise in addition to translations). This is indeed captured by our model (Fig. 3H), with perfect path integration when the uncertainty is low up to 1000 time steps (the discretisation of state space means that uncertainty below 0.075 has no effect ), and monotonically increasing path integration error when the uncertainty is higher.

### 3.1 NEURAL IMPLEMENTATION FOR DEEP REINFORCEMENT LEARNING

Our proposed framework supports prediction and planning under arbitrary direction actions and path integration. To further demonstrate its utility in non-spatial tasks, we propose GridCell-DQN (gc-DQN), a neural network implementation based on a modified version of the classic Deep-Q Network (DQN, Mnih et al. [31]). The architecture of gc-DQN is designed so that quantities corresponding to combinations of Fourier modes weighted by action-dependent values are explicitly available to action-value computation in addition to value estimates based on the state-inputs alone. This should enable the network to predict future state occupancy and thus facilitate the learning of Q values. We evaluated the performance of gc-DQN on the CartPole task (Barto et al. [3]) and compared with plain DQN.

We restrict our introduction of the gc-DQN architecture based on the evaluation on the CartPole task. The state space of the CartPole task is $4$-dimensional, corresponding to the cart position and velocity, pole angle and angular velocity, and there are $2$ possible actions ($0$ and $1$ corresponding to cart movement left or right). The Q-values are learnt using the standard temporal-difference rule (Sutton and Barto [39]). The gc-DQN has 2 additional sub-networks (below the standard DQN in Fig. 4.A): the first takes as inputs the $n$ low-frequency Fourier modes over the state space after discretisation into $8^4$ bins and has $n_{actions} = 2$ outputs to allow representation for each action (left or right) given

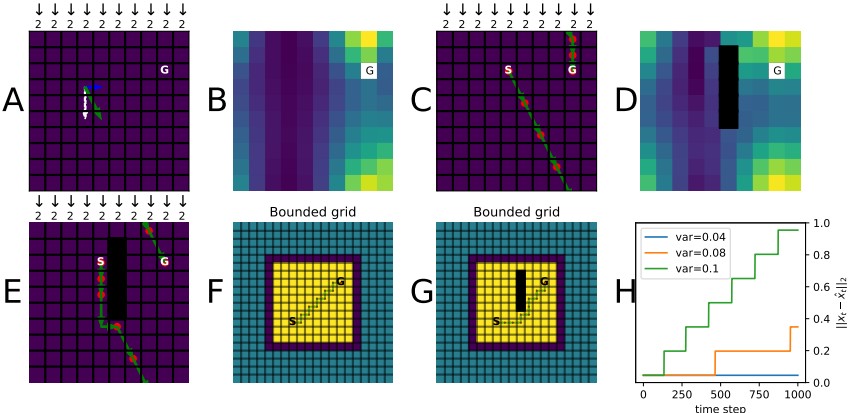

Figure 3: **Application to spatial navigation in grid worlds. A**: The $10 \times 10$ windy grid world task environment, with toroidal boundary conditions and a constant external force causing two units of displacement southward acting on every state (white arrow: wind; blue arrow: one-step action rightward; green arrow: actual displacement, G: goal state); **B**: Estimated SR (using Eq. 2 and Eq. 7) under diffusion plus the wind effect (color indicates the strength of future probabilities of occupying the target states); **C**: Path following the diffusion SR plus the wind effect; **D**: Updated diffusion SR plus wind given the insertion of a barrier (dark blocks); **E**: Path following the updated SR; **F**: Navigation in the task space ($S_0$, yellow) with boundaries (magenta) embedded in a pseudo state space ($S_p$, blue), utilising the Fourier modes computed from $S_p$; **G**: Navigation in $S_0$ with inserted local obstacles (black); **H**: Path integration error over timesteps under different levels of diffusion.

the input Fourier modes; the second takes as inputs the state variables and action and has $2n$ outputs which serve as action-dependent multipliers to the connection weights from the input layer of the first network. The second sub-network receives state as well as action inputs to mitigate the absence of translation-invariance. The outputs of the first network and of the standard DQN are fully connected to an output layer to learn the updated Q values. We also evaluated a model-based version of gc-DQN based on deep Dyna agents (Peng et al. [33]). Preliminary results in Fig. 4(B) show that the gc-DQN and deep gc-Dyna-Q greatly accelerates learning comparing to the baseline agents, with relatively minor increase in the model complexity and computational costs. The results support our hypothesis that Fourier eigenvectors weighted by action-specific values can aid prediction (in this case, of future value). See Appendix. E for details.

The focus of this paper is proposing a theoretical framework for state representation, prediction, planning and path integration via grid-like eigenvectors and action-dependent eigenvalues. The proposed gc-DQN is only a preliminary attempt towards a neural network implementation of the proposed approach (see also Mahadevan and Maggioni [29]), more rigorous studies in this direction is left for future work.

## 4  A UNIFYING FRAMEWORK FOR MODELS OF GRID CELL FIRING

Our focus so far has been on proposing a flexible and efficient extension of the prediction models of grid cells (Baram et al. [2]; Dordek et al. [15]; Stachenfeld et al. [38]) to arbitrary directed transitions. However, many other computational models of grid cells emphasise path integration rather than inputs from place cells, such as continuous attractor network (CAN) models, in which grid-like patterns emerge in recurrently connected networks performing path integration (Fuhs and Touretzky [18]; Burak and Fiete [6]; Corneil and Gerstner [13]), and oscillatory interference (OI) models in which grid-like patterns reflect coincidence detection of velocity-dependent oscillatory inputs during path integration (Hasselmo [22]; Burgess [8]; Welday et al. [42]). Here we build upon the previous work of unifying the prediction and path integration models of grid cells firing by Sorscher et al. [37], to show the equivalence of our generalised prediction framework with the CAN models of grid cells; we additionally show the equivalence between the proposed model and oscilla-

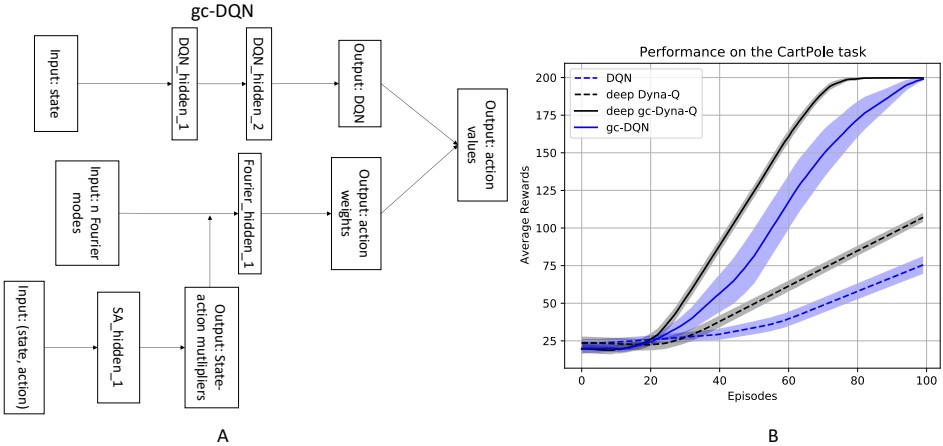

Figure 4: **Neural network implementation of the proposed grid cell model for reinforcement learning. A**: Schematic illustration of the gc-DQN agent. **B**: Evaluation of gc-DQN and the baseline DQN agents in the CartPole task (Barto et al. [3]). The evaluations are computed given 5 random seeds. See Appendix. E for details of the implementation.

tory interference models in terms of their interpretations of path integration and theta phase precession.

### 4.1 RELATION TO CONTINUOUS ATTRACTOR NETWORK MODELS OF GRID CELLS

One of the most prominent unifying analyses for different grid cell models (Sorscher et al. [37]) proves the equivalence between maximising a spatial representation objective function under the prediction models and the pattern formation dynamics of CAN models of path integration. Their analysis, however, does not include non-zero velocity inputs, corresponding to the asymmetric velocity-dependent connectivities in CANs which perform path integration (Fuhs and Touretzky [18]; Burak and Fiete [6]). We can explicitly address this equivalence using our framework. Assuming that grid cell firing rate, $g$, reflects linear combinations of selected Fourier modes (e.g., 6 Fourier modes at $\frac{\pi}{3}$ radians increments with the same frequencies):

$$g = \sum_{j=1}^{G} w_j f_j \tag{9}$$

where $f_j$'s are the selected Fourier modes with weights $w_j$.

Note that our proposed model is on the theory-level, rather than the implementation level of CAN models, hence we do not assume any specific neural network structure here. Utilising the grid cell firing described by Eq. 9, followed by similar analysis as in Sorscher et al. [37], we show that, under non-zero velocities, the differential equations governing the dynamics of the CAN model updates are equivalent to the derivative of the Lagragian equation underlying the optimsation problem of the prediction models (up to scaling factors). Hence we show that under non-stationary transition dynamics, the CAN models and the prediction models should yield identical updates to grid cell firing (up to scaling factors). The complete proof can be found in Appendix B.

### 4.2 RELATION TO OSCILLATORY INTERFERENCE MODELS OF GRID CELLS

Another major class of computational models of grid cells is the oscillatory interference model, here we show the equivalence between the generalised prediction model and the OI models of grid cells by showing that they perform path integration via similar phase coding.

In OI models of grid cells, path integration is achieved via the phases of the "velocity controlled oscillators" (VCOs), which encode movement speed and direction by variations in burst firing frequency. The VCOs generate grid-like firing patterns via coincidence detection by the grid cells

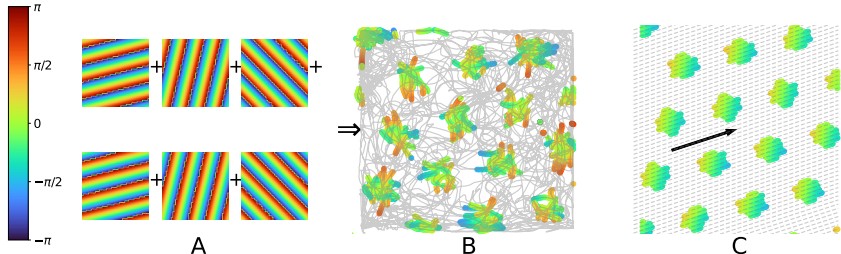

Figure 5: **Generated grid-like patterns given our model**. **A**: 6 input Fourier modes with the same wavelength (complex phases shown); **B**: Simulated grid cell firing given a real rat trajectory (gray line) using coincidence detection and baseline modulation given the 6 Fourier modes inputs in A, each spike is represented by a colored scatter with the color indicates the corresponding "theta" phase at the spiking time; **C**: Simulated grid cell firing fields given multiple runs in the same direction (black arrow) showing theta phase precession.

(Burgess [8]; Hasselmo [22]; Welday et al. [42]). The variation of frequency with velocity produces a phase code for displacement, enabling the modelled grid cells to perform path integration. Namely, VCOs change their frequencies relative to the baseline according to the projection of current velocity, $v(t)$, onto the VCO's "preferred direction", $\mathbf{d}$:

$$f_a(t) = f_b(t) + \beta \mathbf{v}(t) \cdot \mathbf{d} \qquad (10)$$

where $\beta$ is a positive constant, and $f_b(t)$ is the baseline frequency (the $4-11$Hz EEG theta rhythm). It follows that VCOs perform linear path integration since the relative phase between VCO and baseline at time $t$, $\phi_{ab}(t) = \phi_a(t) - \phi_b(t)$, is proportional to the displacement in the preferred direction:

$$\phi_{ab}(t) - \phi_{ab}(0) = \int_0^t 2\pi[f_a(\tau) - f_b(\tau)]d\tau = 2\pi\beta[\mathbf{x}(t) - \mathbf{x}(0)] \cdot \mathbf{d} \qquad (11)$$

where $\mathbf{x}(t)$ is the agent's location at time $t$. The interference of VCOs whose preferred directions differ by multiples of $\pi/3$ generates grid-like patterns, provides an explanation of "theta phase precession" in grid cells (in which firing phase relative to the theta rhythm encodes distance travelled through the firing field; Hafting et al. [21]; Burgess [8]) and complements the attractor dynamics given by symmetrical connections between grid cells (Bush and Burgess [9]). We note that the main experimental results held against OI models (that bats and humans do not have reliable theta frequency oscillations) has recently been resolved: the required phase coding (theta phase precession) can occur relative to a variable baseline frequency (Bush and Burgess [10]) and has now been found in both bats and humans (Eliav et al. [17]; Qasim et al. [35]).

We simulated the firing of a grid cell with 6 Fourier mode inputs (Fig. 5 A-B), each firing a spike at its complex phase in the current state, as a leaky integrate and fire neuron performing coincidence detection, using a real trajectory of a rat exploring a $50cm \times 50cm$ box. At each time step (corresponding to one theta cycle), the phase of each Fourier mode is updated according to Eq. 7 given the current velocity, and fires a spike at this phase if it is within the interval $[-\pi/4, \pi/4]$ (modelling modulation by the baseline theta oscillation). The grid cell fires a spike at the current location if the integrated inputs reach a threshold. Note that we could also simulate a set of grid cells, with different offsets (depending on the initial phases of the Fourier modes) and different scales and orientations (depending on the choice of Fourier modes), such that the grid cells, like the Fourier modes, comprise a basis for the state space and do so on the basis of path integration (for which environmental inputs are only required to prevent error accumulation [8; 7]).

Grid cells show "precession" in their firing phase relative to theta as the animal moves through the firing field (signaling distance travelled within the field; Hafting et al. [21]; Jeewajee et al. [23]; Climer et al. [12]). Our model captures this, like an OI model. The Fourier modes whose wavevectors that are aligned with the current direction of translation advance in phase as the movement progresses. Phase precession results from assuming that the Fourier modes aligned with movement direction are the dominant influence on grid cell firing (c.f. those aligned to the reverse direction). The baseline "theta frequency" corresponds to the mean rate of change of phase of all Fourier components and so could vary (e.g., for noise reduction, see Burgess and Burgess [7]; Burgess [8]), without precluding

phase coding (Eliav et al. [17]; Bush and Burgess [10]). By simulating straight runs, we can see clear late-to-early phase precession (Fig. 5 C), as observed in grid cells.

Thus, the OI model and our model perform path integration or prediction in the same way: the phase of each VCO changes corresponding to the component of translation along the VCO's preferred direction, which is exactly analogous to the complex phases of the Fourier modes being updated to reflect transitions along their wavevectors (multiplication by corresponding eigenvalues, Eq. 7).

## 5 DISCUSSION

Understanding how different actions affect the agent's state across environments is essential for generalisation. Existing models are capable of such prediction under a single fixed transition matrix, e.g. corresponding to symmetrical diffusion, by using eigenvectors of this transition matrix as a basis for representing state occupancy (Baram et al. [2]; Corneil and Gerstner [13]; Stachenfeld et al. [38]). Here we generalised these models to provide a mathematical framework for predicting the effects of specific actions, so long as their effects (and corresponding transition matrices) are translation invariant. This uses a common set of eigenvectors of all such matrices (Fourier modes of the state space) to represent state occupancy, so that the effects of actions correspond to multiplication by action-specific eigenvectors.

This model explains how grid cells (as superpositions of Fourier modes) could support prediction of the effects of actions across environments that share the same underlying transition structure (see also Whittington et al. [43]), and could, for example, perform sense-of-direction planning in new environments (i.e., finding combinations of actions that most likely lead to the target state by ignoring local obstacles). We assume that other (e.g., fronto-parietal) brain areas are responsible for detecting and avoiding obstacles following the overall direction provided by the grid cells [16; 28]. However, topology-dependent modifications to grid firing patterns could be used to accommodate local deviations from translation-invariance (e.g. obstacles), utilising the Woodbury inversion formulae, see Fig. 3E, Appendix A3 and Piray and Daw [34]. We also show show our framework corresponds to other computational models of grid cells based on path integration, and provide a functional explanation for theta phase precession.

A number of questions and predictions are raised by the proposed model. If a basis of neurons with Fourier-mode-like firing patterns act as inputs to cells in entorhinal cortex, then grid cell firing patterns are only a small proportion of the set of firing patterns that could be synthesised. This is consistent with the existence of periodic non-grid cells in entorhinal cortex that resemble combinations of small numbers of Fourier modes (Krupic et al. [27]). We predict the use of the same set of grid cells (superpositions of Fourier modes inputs) for indicating "sense of direction" to goal locations across different environments after a single visit, i.e. showing generalisation across Euclidean environments. Finally, the proposed model predicts future state occupancy from the transition matrix, future work could also consider the reversed direction: inferring the translation between two locations given the phase codes for each (see Bush et al. [11]), as linked discriminative and generative models.

The current model applies to translation-invariant transition structures, and use of the Fourier shift theorem to calculate eigenvalues also assumes a Euclidean state space. We demonstrated ways of generalising planning to bounded or locally non-translation invariant transition structures in Section 3. We note that machine learning methods based on a similar premise (creating a single representation to support planning via multiple different actions) might work even when the transitions are not strictly translation invariant (e.g., family trees, see Whittington et al. [43]). Here we showed that, by giving standard DQN agents the ability to represent the current state as eigenvectors of the state space weighted by state- and action-specific values, significantly improves learning of the CartPole task (Fig. 4), which is not strictly spatial or translation invariant. Hence our approach offers potential generalisation to non-spatial tasks, such as transitive inference (Von Fersen et al. [41], Appendix D). Future work will involve more rigorous study of the neural network implementation of the proposed grid cell model and its application to reinforcement learning. A future direction in generalising the current model to non-spatial tasks will be to consider Fourier analysis on groups of operators utilising group-theoretic knowledge (Kondor and Trivedi [25]; Gao et al. [19]).

ACKNOWLEDGEMENTS

C.Y. thanks a DeepMind PhD studentship offered by the UCL Centre for Artificial Intelligence. T.E.J.B and N.B. thank the Wellcome Trust for support. The authors would like to thank Daniel Bush, Talfan Evans, Kimberly Stachenfeld, Will de Cothi and Maneesh Sahani for helpful comments and discussions. The authors declare no competing financial interests.

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

# A    SOME PROOFS IN SECTIONS 3

**Proposition A.1** *Given our assumption of periodic boundary condition, the transition matrix, $T \in \mathbb{C}^{N \times N}$ (Eq. 3), is indeed a circulant matrix.*

*Proof*    It is easy to see that the proposition holds trivially for transition matrices with only one-step translations but without Gaussian spread. Hence here we only show the proof for the case where the transition structure includes both Gaussian spread and one-step translations.

Consider for an arbitrary transition matrix $T$ for a 2D rectangular environment with length $L$ and width $W$, and the underlying transition velocity is $v = (v_x, v_y)$, remembering that $\mathbf{T}$ is the $LW \times LW$ 2D matrix formed from concatenating rows from what would be the 4D matrix of transitions between all pairs of states in a $L \times W$ 2D state space. An arbitrary entry on the $k$th lower subdiagonal is $T_{i,i-k} = \mathbb{P}(x(t+1) = i - k | x(t) = i)$ for any suitable state $i$ given $k$ (i.e., $i \geq k$). If the Gaussian spread is radially symmetric with constant variance across states, the value of $T_{i,i-k}$ only depends on the distance between state $i - k$ and the state $i_v$, where $i_v$ is the translated state of state $i$ given the effect of the velocity $v$. The states $i - k$ and $i_v$ are equivalent to the states $((i-k)//L, (i-k) \bmod L)$ and $(i//L + v_x, i \bmod L + v_y)$ in the two-dimensional spatial domain respectively (where $a//b$ denotes the integer part of $a/b$). Note that we need to have the velocity $v \in [\pm L/2, \pm W/2]$ so that the translation leaves the actual distance $d$ unchanged. The distance between the state $i - k$ and the expected state $i_v$ in the 2D state space is then

$$d = \sqrt{((i-k)//L - i//L - v_x)^2 + ((i-k) \bmod L - i \bmod L - v_y)^2} \tag{12}$$

For any arbitrary $i' \neq i$ such that $i' = i + m$, we could compute similarly the distance between state $i' - k$ and its corresponding expected state (Gaussian center) $i' + v$. After some algebra, we have that the distance between states $i - k$ and $i + v$ equals the distance between states $i' - k$ and $i' + v$.

$$d' = \sqrt{((i'-k)//L - i'//L - v_x)^2 + ((i'-k) \bmod L - i' \bmod L - v_y)^2}$$
$$= \sqrt{((i+\delta-k)//L - (i+\delta)//L - v_x)^2 + ((i+\delta-k) \bmod L - (i+\delta) \bmod L - v_y)^2} \tag{13}$$

Now if we look at the two square terms within the square root separately, we have

$$((i+\delta-k)//L - (i+\delta)//L - v_x)^2$$
$$= ((i-k)//L + \delta//L - i//L - \delta//L - v_x)^2 \tag{14}$$
$$= ((i-k)//L - i//L - v_x)^2$$

$$((i+\delta-k) \bmod L - (i+\delta) \bmod L - v_y)^2$$
$$= (((i-k) \bmod L + \delta \bmod L) \bmod L - (i \bmod L + \delta \bmod L) \bmod L - v_y)^2$$
$$= ((i-k) \bmod L + \delta \bmod L - i \bmod L - \delta \bmod L - v_y)^2 \tag{15}$$
$$= ((i-k) \bmod L - i \bmod L - v_y)^2$$

The second equality holds due to the fact that $(i - k) \bmod L + \delta \bmod L \leq L$ since this is simply the $x$-position of state $i + \delta - k$, which is never larger than $L$, hence $((i-k) \bmod L + \delta \bmod L) \bmod L = (i-k) \bmod L + \delta \bmod L$. Similarly $(i \bmod L + \delta \bmod L) \bmod L = i \bmod L + \delta \bmod L$. Hence we have

$$d' = \sqrt{((i'-k)//L - i'//L - v_x)^2 + ((i'-k) \bmod L - i' \bmod L - v_y)^2} = d \tag{16}$$

Hence all entries on the $k$th lower subdiagonal are identical, i.e. $T_{i,i-k} = T_{i',i'-k}$ for all $1 \leq k \leq LW - 1$. And by similar arguments, we could show that all entries on the $k$th upper subdiagonal are identical (for $1 \leq k \leq LW - 1$), and equals to the corresponding entries on the $LW - k$th lower subdiagonals. And the fact that all the main diagonal entries are identical is immediate from the problem setting. Hence our target transition matrix is indeed a circulant matrix. (Note that in simulations the transition matrix will only be approximately circulant due to normalisation and numerical issues.)

Now we consider the corresponding $(LW - k)$th upper subdiagonal (to the $k$th lower subdiagonal), by similar arguments, we have that for any $T_{i'',i''+k} = \mathbb{P}(x(t+1) = i'' + k | x(t) = i'')$ for suitable $i''$ (i.e. $i'' + k \le L$), the distance between the state $i'' + k$ and expected next state $i'' + v_t$ are the same as $d$, which is equivalent to $T_{i'',i''+k} = T_{i,k-i}$. Hence all entries on the $(LW - k)$th upper subdiagonal are identical and equal to the entries on the $k$th lower subdiagonal. This holds for arbitrary $1 \le k \le LW - 1$. $\qquad\square$

**Proposition A.2** *For any circulant matrix $T \in \mathbb{C}^{N \times N}$ as shown in Eq. 3, its $k$th eigenvector takes the form:*

$$\mathbf{v}^k = \frac{1}{\sqrt{N}} \left[ 1, \omega_k, \omega_k^2, \cdots, \omega_k^{N-1} \right]^T \tag{17}$$

*where $\omega_k = \exp\left(\frac{2\pi k i}{N}\right)$ is the $k$th $N$th root of unity, and the set of eigenvalues equals to the set of DFTs of an arbitrary row/column of $T$.*

*Proof*  Firstly, note that the product between the circulant matrix $T$ and an arbitrary vector $\mathbf{v}$ is equivalent to a convolution.

$$\mathbf{w} = T \cdot \mathbf{v} = \begin{bmatrix} T_0 & T_{N-1} & \cdots & T_2 & T_1 \\ T_1 & T_0 & T_{N-1} & \cdots & T_2 \\ \vdots & T_1 & T_0 & \ddots & \vdots \\ T_{N-2} & \cdots & \ddots & \ddots & T_{N-1} \\ T_{N-1} & T_{N-2} & \cdots & T_1 & T_0 \end{bmatrix} \cdot \begin{bmatrix} v_0 \\ v_1 \\ \vdots \\ v_{N-1} \end{bmatrix} \tag{18}$$

And we immediately have that

$$w_k = \sum_{j=0}^{N-1} T_{j-k} v_j \tag{19}$$

This is true due to the periodicity of the entries given by the circulant structure. Then if we take the dot product of $T$ and an arbitrary vector $\mathbf{v}^m$ of the form shown in Eq. 17, the $l$th entry of the output vector has the following form.

$$\sum_{j=0}^{N-1} T_{j-l} \omega_j^m = \omega_l^m \sum_{j=0}^{N-1} T_{j-l} \omega_{j-l}^m \tag{20}$$

where the equality holds since $\omega_j^m = \exp\left(\frac{2\pi i}{N} jm\right) = \exp\left(\frac{2\pi i}{N}(j-l)m\right)\exp\left(\frac{2\pi i}{N}lm\right) = \omega_l^m \omega_{j-l}^m$. Note that the last sum in Eq. 20 is independent of the choice of $l$ since both $T_j$ and $\omega_j$ are periodic hence any change in $l$ is simply rearranging the terms in the summation. Also we have that $\omega_l^m = \omega_m^l$ is the $l$th entry of the eigenvector $v_m$. Hence we have

$$T\mathbf{v}_m = \lambda_m \mathbf{v}_m \tag{21}$$

where

$$\lambda_m = \sum_{j=0}^{N-1} T_j \omega_j^m \tag{22}$$

for $m = 0, \ldots, N-1$. Hence for an arbitrary $N \times N$ circulant matrix $T$, the eigenvalues take the form as shown in Eq. 22 and the corresponding eigenvectors take the form as shown in Eq. 17, and the eigenvalues are equivalent to the DFT of the first row of the circulant matrix immediately follows from Eq. 22 and the definition of DFT (Bracewell [4]). $\qquad\square$

The predicted phase change in the eigenvalues over the eigenvalues of the baseline symmetric transition matrix computed with Fourier modes computed via Fourier shift theorem (Eq. 7) under our formulation perfectly captures the actual phase changes caused by the one-step translations in the eigenvalues between the symmetric and and asymmetric transition matrices, as shown in Fig. 6A. However, when the transition dynamics is a combination of diffusion and one-step translations, the predicted phase changes in eigenvalues will no longer perfectly match the actual phase changes observed as shown in Fig. 6B, and the oscillation is caused by the diffusion process. Namely, although the expected translation is indicated by the velocity, the actual translation spans a range of states depending on the width of the diffusion field.

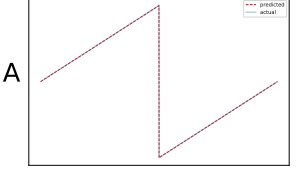 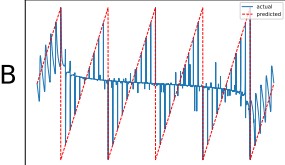

Figure 6: **Application of Fourier shift theorem for predicting changes in eigenvalues**. We show the ground-truth (blue) and predicted (red) phase shifts of the eigenvalues of transition matrices given arbitrary drift velocity for: **A**: plain translation (5 units rightward); **(B)**: diffusion with one-step translations (5 states rightward + diffusion). The horizontal and vertical axes represent the indices of the eigenvalues and the corresponding phase changes (in radians) respectively.

**Proposition A.3** *The updated SR given the insertion of a barrier is*

$$S = S_0 - C(I + RC)^{-1}RS_0 \tag{23}$$

*where $S_0$ and $S$ are the initial and updated SR, $R = S_0[J, :]$ and $C = S_0[:, J]$ are the J-th rows and columns of $S_0$ respectively, where $J$ is the index set of states adjacent to the inserted barrier.*

*Proof* This derivation is inspired by Piray and Daw [34]. Given the definition of the SR, we have

$$S = (I - \gamma T)^{-1}, \quad S_0 = (I - \gamma T_0)^{-1} \in \mathbf{R}^{N \times N} \tag{24}$$

where $N$ is the number of states. Given the insertion of a barrier, $S$ and $S_0$ only differ in their $j$-th rows for $j \in J$ where $J$ is the index set of states adjacent to the barrier. Hence we could write:

$$R = T[J, :] - T_0[J, :] \in \mathbb{R}^{|J| \times N} \tag{25}$$

Then if we have $E \in \mathbb{R}^{|J| \times N}$ with zeros everywhere but ones on the $j$-th rows for $j \in J$, then by setting $W = I - T$ and $W_0 = I - T_0$, we could write:

$$W = W_0 + ER \tag{26}$$

The Woodbury inversion formula is usually use in cases whn we are trying to compute the inverse of a matrix given a low-dimensional perturbution (Riedel [36]).

$$(A + UCV)^{-1} = A^{-1} - A^{-1}U(C^{-1} + VA^{-1}U)^{-1}VA^{-1} \tag{27}$$

Hence by applying the Woodbury inversion formula, we have:

$$W^{-1} = W_0^{-1} - EW_0^{-1}(I + REW_0^{-1})^{-1}RW_0^{-1}$$
$$\Rightarrow S = S_0 - C(I + RC)^{-1}RS_0 \tag{28}$$

where $C = ES_0$ are the $j$-th columns of $S_0$ for $j \in J$. □

**Proposition A.4** *The "sense of direction", $\theta^*$, is given by the form shown in Eq. 8.*

*Proof* Essentially, we wish to find value of $\theta$ such that under the drift velocity $\mathbf{v}_\theta = (v\cos(\theta), v\sin(\theta))$, given the start and target states, $\mathbf{s}_0$ and $\mathbf{s}_G$, the future discounted occupancy of $\mathbf{s}_G$ starting from $\mathbf{s}_0$ (or $W[\mathbf{s}_0, \mathbf{s}_G]$, where $W$ is the SR matrix), is maximised. Under our formulation, $W$ can be calculated as follows:

$$W = F\text{diag}(1/(1 - \gamma\Lambda^{\mathbf{v}_\theta}))F^{-1} \tag{29}$$

where $F$ is the DFT matrix (Eq. 4), and $\Lambda^{\mathbf{v}_\theta}$ is the set of eigenvalues of the transition matrix given velocity $\mathbf{v}_\theta$. From our analysis based on Fourier shift theorem (Eq. 7), for each $\lambda_i^{\mathbf{v}_\theta} \in \Lambda^{\mathbf{v}_\theta}$, we have that:

$$\lambda_i^{\mathbf{v}_\theta} = D_i\omega^{\mathbf{v}_\theta \cdot \mathbf{k}_i} \tag{30}$$

where $D_i$ is the $i$th eigenvalue of the symmetric (baseline) diffusion transition matrix, and $\mathbf{k}_i$ is the wavevector for the $i$th Fourier mode. Then using linear algebra, we immediately arrive at the expression in Eq. 8. □

# B  SOME PROOFS IN SECTION 4

**Proposition B.1** *The equations governing the dynamics of the prediction model and the CAN model of path integration are equivalent.*

*Proof*    We show the proof under the single-cell formulation, which can be immediately generalised to the situation with multiple cells.

We firstly note that the prediction model can be mathematically categorised as minimising the following reconstruction objective function.

$$\mathcal{E}(g) = ||\mathbf{T} - gw||_F^2 \tag{31}$$

where $g \in \mathbb{R}^{n_G}$ represents the grid cells firing rates, and $w \in \mathbb{R}^{1 \times N}$ represents the linear readout weights. Following Sorscher et al. [37], we replace $w$ by its optimal value given a fixed $g$, i.e., $\hat{w} = (g^T g)^{-1} g^T \mathbf{T}$. Note that any scaling of $g$ can be absorbed into a corresponding reversed scaling into $\hat{w}$, hence $g$ is assumed to be of unit modulus (or the matrix $G$ can be taken to be orthonormal in the multi-cell case). Additionally, following the non-negativity constraint proposed in Dordek et al. [15], the overall optimisation problem becomes.

$$\min \mathcal{E}(g) = ||\mathbf{T} - g\hat{w}||_F^2, \text{ subject to } g^T g = 1, \text{ and } g_i >= 0 \forall i \tag{32}$$

Hence we can immediately write down the Lagrangian as follows.

$$\mathcal{L} = g^T \mathbf{T}^0 g - \gamma g^T g + \mu \mathbb{1}^T g \tag{33}$$

where $\gamma$ and $\mu$ are the multiplicative constant for the additive penalty terms corresponding to the constraints in Eq. 32. The derivate of the Lagrangian with respect to $g$ then takes the following form.

$$\frac{dg}{dt} = \begin{cases} -\gamma g + Tg + \mu, & g > 0 \\ -\gamma g + \sigma(Tg + \mu), & g = 0 \end{cases} \tag{34}$$

where $\sigma(\cdot)$ is the rectified linear function. Inserting the grid cell firing representation as a linear summation of the Fourier modes into Eq. 34, we obtain the following.

$$\frac{dg}{dt} = \begin{cases} -\alpha g + \gamma(\sum_{j=1}^G \lambda_j w_j f_j) + \mu, & g > 0 \\ -\alpha g + \sigma(\gamma(\sum_{j=1}^G \lambda_j w_j f_j) + \mu), & g = 0 \end{cases} \tag{35}$$

where $\lambda_j$ are the corresponding eigenvalue of $f_j$ with respect to the (symmetric) transition matrix, $\mathbf{T}^0$.

The dynamics of the grid cells under the CAN models can be written as following.

$$\tau \frac{dg}{dt} = -g + \sigma(\mathbf{W}g + b(v)) \tag{36}$$

where $\mathbf{W}$ is the recurrent connectivity matrix, $b(v)$ is the velocity-dependent feedforward input to the grid cell under the CAN model which involves a constant baseline term and a velocity dependent term (Burak and Fiete [6]).

Now suppose the agent is moving under non-zero velocity, $\mathbf{v}$. Given the grid cell firing represented by the linear summation of Fourier modes (Eq. 9), Eq. 35 can be written as following,

$$\frac{dg}{dt} = \begin{cases} -\alpha g + \frac{2\pi i}{N} \gamma \mathbf{T}^0 g + (\frac{2\pi i}{N} \gamma \sum_{j=1}^G \lambda_j w_j f_j(\langle v, \hat{\mathbf{e}}_j \rangle - 1) + \mu), & g > 0 \\ -\alpha g + \frac{2\pi i}{N} \gamma \mathbf{T}^0 g + \sigma(\frac{2\pi i}{N} \gamma \sum_{j=1}^G \lambda_j w_j f_j(\langle v, \hat{\mathbf{e}}_j \rangle - 1) + \mu), & g = 0 \end{cases} \tag{37}$$

where $\mathbf{e}_j$ is the unit-norm wavevector of the Fourier mode $f_j$ for all $j$.

Now check with Eq. 36 by setting

$$\tau = \frac{1}{\alpha},$$
$$\mathbf{W} = \frac{2\pi i}{N} \frac{\gamma}{\alpha} \mathbf{T}^0,$$
$$b(v) = (\frac{2\pi i}{N} \gamma \sum_{j=1}^G \lambda_j w_j f_j(\langle v, \hat{\mathbf{e}}_j \rangle - 1) + \mu)/\alpha, \tag{38}$$

By checking that when $g > 0$, $\sigma(\frac{2\pi i}{N}\gamma \mathbf{T}^0 g + \sigma(\frac{2\pi i}{N}\gamma \sum_{j=1}^{G} \lambda_j w_j f_j(\langle v, \hat{\mathbf{e}}_j \rangle - 1) + \mu)) = \frac{2\pi i}{N}\gamma \mathbf{T}^0 g + \sigma(\frac{2\pi i}{N}\gamma \sum_{j=1}^{G} \lambda_j w_j f_j(\langle v, \hat{\mathbf{e}}_j \rangle - 1) + \mu)$, we see that under non-zero velocity inputs, by appropriately adjusting the additive velocity input term, $b(v)$, the equations governing the dynamics for the normative and mechanistic models are equivalent. $\qquad\square$

## C  2D FOURIER MODES

We know that the Fourier basis vectors from Eq. 4 form plane waves as shown in Fig. 5. From standard Fourier analysis in 2D space, the 2D Fourier modes form an orthonormal basis, and takes the following form.

$$\mathbf{v}_{\mathbf{u}}[\mathbf{x}] = \exp\left(2\pi i \mathbf{u} \cdot \mathbf{x}\right) \qquad (39)$$

where the 2D Fourier basis vectors are encoded by the position vectors $\mathbf{u} = (u_1/L, u_2/W) \in [0,1] \times [0,1]$ (position vectors of each location in the $L \times W$ environment projected onto $[0,1] \times [0,1]$). The direction of the encoder position vector $\mathbf{u}$ represents the direction of the plane wave and the frequency of the plane wave is the unnormalised direction vector $||\mathbf{u}'||$ (where $\mathbf{u}' = \mathbf{u} \times (L, W)$), note that $\mathbf{u}'$ is also the wavevector for the plane wave. This is a slightly different formulation comparing to the formulation given in Eq. 4, which consider the state space as a 1-dimensional flattened vector of the 2-dimensional environment, hence the Fourier basis vectors are the corresponding 1-dimensional Fourier modes. Though both formulation give us the same set of Fourier basis vectors, under the definition in Eq. 39, we could easily track the frequency and direction of the plane wave formed from the 2D Fourier modes. And the phase shift via the Fourier shift theorem 7 equivalently applies for this 2D Fourier formulation.

The Fourier modes comprises a basis for representing any distribution over the task state space, so we could use a linearly weighted combination of Fourier modes to reconstruct any firing patterns, such as those observed in place cells (Welday et al. [42], Fig. 8). However note that the coincidence detection of small number of oscillators with different frequencies will generate periodic patterns, e.g., grid cells, and more oscillators will be needed for those with more local firing fields such as place cells. Note that the total number of Fourier modes equals the number of states in the environment (e.g., $LW$ for the $L \times W$ rectangular environment on a square grid), and it could be infeasible and inefficient to compute and store a large number of such Fourier modes (or neurons with VCO-like firing patterns) in the brain. Hence here we only use the principal modes (taking the top $n$ Fourier basis vectors in terms of the corresponding eigenvalues (frequencies)), within contain the majority of the information is contained, with the number of principal modes depending on the desired reconstruction resolution. We utilised the top 100 principal Fourier modes for most of the simulations in the main text (see Fig. 7 for a typical fixed set of Fourier modes). Fig. 8 demonstrates that the small number of Fourier modes are able to reconstruct grid cells firing fields with various spacings and orientations, and place cells firing fields.

## D  TRANSITIVE INFERENCE

In the main paper we argued that the same set of eigenvectors can be used to predict future occupancy distribution given the transition matrix for symmetrical relations like diffusion between adjacent states and directed transitions (e.g. moving N S E W). Here we briefly discuss the generalisation of our model to non-spatial tasks.

We could apply our method to the one-dimensional transitive inference tasks of this type. e.g., given $A > B$, $B > C$, $C > D$, then infer if $A > D$ (Von Fersen et al. [41])? This would be like having a 1D track (and Fourier eigenvectors for 1-step transitions in both directions) corresponding to actions "greater" or "smaller", and using "intuitive planning" to see if using eigenvalues for "greater" will take you from A to B in the discounted future more likely than eigenvalues for "smaller". In order to deal with the non-periodicity of the task, we simulate transitive inference in a small subset of the state space of the torus. As shown in Fig. 9, we see that our framework correctly predicts the transitive relationship between the chosen state $x_{129}$ and states close to $x_{129}$.

Despite the simplicity of 1D transitive inference (Fig. 9), our model is still an advance on the original intuitive planning method in being able to predict the effects of both "greater" and "smaller" transitions

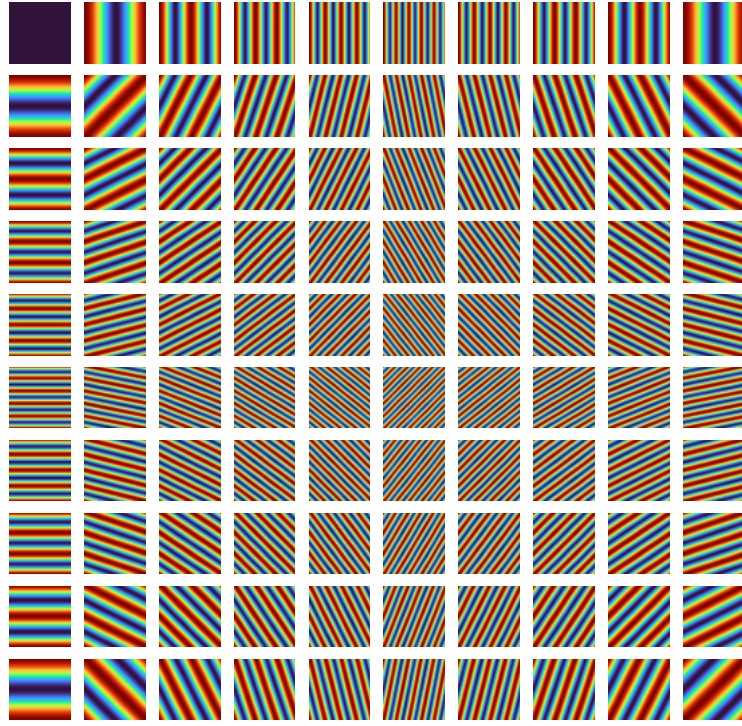

Figure 7: Phase plots of 100 chosen low-frequency Fourier basis vectors with different frequencies and wavevectors.

with the same set of eigenvectors, rather than being restricted to prediction with one or the other alone.

# E    SIMULATIONS

## E.1    FURTHER DETAILS OF THE GC-DQN AGENT

The overall architecture can be found in the graphical illustration in fig. 4. At each timestep, the state values and a specific action value are fed into a neural network for all possible values of actions (Blue box in the bottom left of fig. 4), which outputs $n_{actions}$ output, where $n$ is the number of Fourier modes inputs to the second network. The output can considered as the specific updates to each Fourier modes corresponding to the action in the current location, like the phase shift in the Fourier shift theorem.

The inputs to the grid cell network are the first $n$ Fourier modes, whose dimensions $(D)$ are determined by the size of the state space. When the state variables are continuous, we compute an approximate size of the state space by discretising each state variable. The number of principal Fourier modes $(n)$ is chosen arbitrarily as long as the majority of the information can be reconstructed from the chosen set of Fourier modes. Higher values of $n$ leads to finer details of the prediction, but also induces higher computational costs.

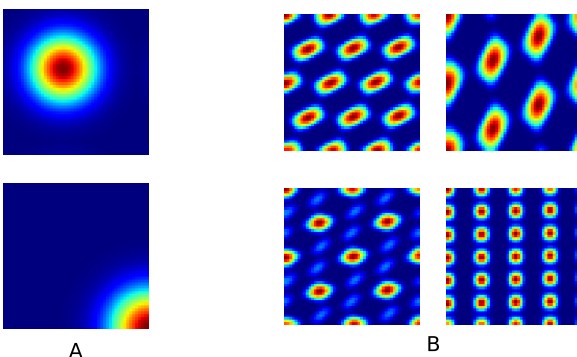

Figure 8: **Constructed place cell and grid cell firing fields from Fourier modes.** **A**: Place cells firing fields constructed from coincidence detection of selected input Fourier modes (bottom plot shows a place field restricted to a small subset of the toroidal state space); **B**: Grid cell firing fields with various spacings and orientations constructed from principal Fourier modes (Fig. 7).

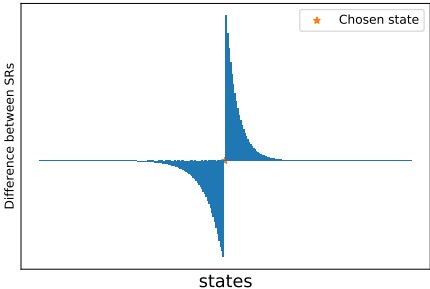

Figure 9: **Generalisation of flexible planning on transitive inference task**. Given $\{x_i\}_{i=0}^{259}$ such that $x_i > x_{i+1}$ for all $i$ (and $x_{259} > x_0$ for ensuring the circulant structure). The bar plot shows that we can correctly infer transitive relations between the chosen state $x_{129}$ (red star) and nearby target states via computing the difference between the discounted future occupancy of the target state under the action-dependent SRs (Eq. 2) corresponding to the "smaller" (left) and "greater" (right) actions. The $x$-axis denotes the states, and the $y$-axis denotes the difference between the SRs.

At each timestep, the $n$ Fourier modes is fed as the input to the grid cell network (shown in the middle row of fig. 4(A)). Each action multiplier (outputs from the state-action network) is multiplied with the corresponding column of the weight matrix between the input layer and the hidden layer of the grid cell network. The outputs of the hidden layer is then transposed, and forward propagate to the output layer of the second network. The computations of the grid cell network is considered to be equivalent to using the Fourier modes to construct a weight value for choosing each action at a given state that aids navigation/planning.

The outputs from the grid cell network and the standard DQN agent is then combined to output a vector, that acts as the values for each action that guides action choice in the current timestep.

### E.2 SIMULATION DETAILS

All simulations were implemented in Python. The simulation details for each task is as follows:

- Fig. 1: The state space is assumed to be a $1D$ ring with 20 states, with the transition probabilities $\mathbb{P}(s_{t+1} = i + 1|s_t = i) = \mathbb{P}(s_{t+1} = i - 1|s_t = i) = 0.5$, and discounting factor $\gamma = 0.9$ for generating the resolvent (Eq. 2).

- Fig. 2: Variance of each (Gaussian) firing field (representing the strength of diffusion) is 3; **B**: $(0, 5)$ drift velocity with increasing diffusion (variance increase by 3 per step); **C**: $(3, 3)$ drift velocity with 0 diffusion; **E**: The successor representation is computed using the Fourier modes and corresponding eigenvalues, with the discounting factor $\gamma = 0.9$.

- Fig. 3: **A**: The wind effect causes $(0, 2)$ (2 units southward) displacement at each timestep; **B, D**: The successor representation is computed given a transition matrix that assumes the variance at each (Gaussian) firing field is $1.5$, followed by directed actions under the wind effects (with 0 diffusion), the discounting factor is $\gamma = 0.9$; **F, G**: The optimal following the ascending values of the successor representation, without any wind effect. The SR is computed given a transition matrix that assumes the variance at each (Gaussian) firing field is $1.5$, followed by directed actions, the discounting factor is $\gamma = 0.9$; All computations are done by working directly with the Fourier modes instead of the transition matrices.

- Fig. 4: The environment is the CartPole task (Barto et al. [3]), and is simulated using the OpenAI gym environment (Brockman et al. [5]). The state value consists of 4 variables: (Cart position, Cart velocity, Pole angle, Pole angular velocity), the action value is an integer takes value from $\{0, 1\}$, where 0 represents moving left, and 1 represents moving right. For constructing the Fourier modes, we discretised each state variable into 8 bins, hence resulting in $8^4$ number of states, and we chose the top 50 low-frequency Fourier modes as the inputs to the grid cell network. The standard DQN agent consists of two fully connected hidden layers with standard ReLU activations, with 48 and 24 units, respectively. The target network is updated every 500 timesteps. The deep Dyna-Q agent is a simplified version of the model proposed in Peng et al. [33], with an additional 2-layer neural network learning the environmental transition dynamics, with 64 and 32 units in each hidden layer followed by ReLU activations. At each timestep, the learnt environment model is called to generate $K$ imaginary trajectories that are used for model-based updates to the DQN agent. The number of model-based updates, $K$, is taken to be 2. The state-action network in the gc-DQN has one hidden layer, with 32 units followed by ReLu activation. The grid cell network has one hidden layers, with hidden size $(n, A)$ followed by ReLU activation, where $n$ represents the number of input Fourier modes, and $A$ represents the number of possible actions. The deep gc-Dyna-Q has similar architecture as the gc-DQN agent, but with an additional environment network that learns the transition dynamics of the environment that is used for model-based updates (with same architecture as the standard deep Dyna-Q agent). All models are learnt using the mean squared error loss function and Adam optimiser (Kingma and Ba [24]) with learning rate $0.001$ and no learning rate decay. The exploration strength, $\epsilon$, is set to be $0.8$ at the start of each independent run, decreases by $0.05$ at each episode, and is bounded below by $0.01$. A total of 5 independent runs of 100 episodes are performed for each agent. Note that 100 episodes were simulated for each independent run due to the limited time and computational resources, but the results show that it is sufficient for demonstrating the increase in performance of the gc-DQN agent comparing to the baseline agents. We will, upon acceptance, show simulations with more episodes (up to the points where convergence

of the baseline agents are observed) in the camera-ready version. All implementations are performed in the TensorFlow framework (Abadi et al. [1]).

- Fig. 5: A: wavevectors of chosen input Fourier modes: $\mathbf{k}_1 = (4/50, 1/50), \mathbf{k}_2 = (1/50, 4/50), \mathbf{k}_3 = (3/50, -3/50), \mathbf{k}_4 = (-4/50, -1/50), \mathbf{k}_5 = (-1/50, -4/50), \mathbf{k}_6 = (-3/50, 3/50)$; B: real rat trajectory projected onto $50 \times 50$ 2D spatial domain, firing phase interval of the input Fourier modes: $[-2.5\pi/12, 2.5\pi/12]$, integration time interval: 8, exponential decay rate: 0.2, grid cell firing threshold: 2.95, directional bias: within $\pm\pi/2$ of the head direction (the range of the relative difference between the direction of the wavevector and the head direction, within which the Fourier modes are allowed to fire); C: running direction: $arctan(1/3)$.

- Fig. 9: The discounting factor: $\gamma = 0.3$, the number of states: 26, number of effective transitive inference states: 10.

The Python-based implementations can be found at `https://github.com/ucabcy/Prediction_and_Generalisation_over_Directed_Actions_by_Grid_Cells`.

