# OpenReview forum: "Prediction and generalisation over directed actions by grid cells"
_ICLR.cc/2021/Conference — ICLR 2021 Poster_

### Official Review · AnonReviewer2 · 2020-10-28
**Efficient method to perform prediction over grid cells.**

**Rating:** 5
**Confidence:** 1

**Review:**

The authors propose an efficient method to predict directed transitions in spatial tasks by extending eigenbasis based prediction model. The authors show equivalence of the proposed method to classical models of path integration by grid cells - continuous attractor networks and oscillatory inferencce.


Strengths:

1) The proposed method is efficient and doesn't need a decomposition for future state occupancy and can be done directly by re-weighing of eigenvector (by eigenvalues) of the transition matrix.

Weaknesses:

1) Lack of quantitative evaluation and comparison with naive baselines makes it hard to grasp the contribution and value added by the proposed method.
2) The datasets used for evaluation (and studying different properties of the proposed method) are oversimplistic.
3) Assumptions made about applicability of the proposed method to tasks without periodic boundary conditions are not substantiated with experimental evidence.

---

### Official Review · AnonReviewer1 · 2020-10-28
**A solid contribution to grid cell modeling - but the paper could be much clearer for a non-specialized audience**

**Rating:** 7
**Confidence:** 4

**Review:**

This paper shows how SR representation theory can be used in a model of grid cells to plan and navigate towards a target. A key insight of the paper is that velocity instructions modify the eigenvalues of the SR but not its eigenvectors, so the eigendecomposition does not need to be recomputed for all candidate velocity instructions, and the eigenbasis can be hard-coded in the neural circuit (by the grid cells).

Strong points:
- important contribution to our understanding of how grid cells could support navigation towards a goal
- rigorous mathematical derivation of the results

Weak points:
- the paper is written for a highly specialized audience, which limits its impact (see below for detailed comments).
- it is unclear to me how the successive exploration of possible directions could be implemented in practice by the grid cell network (see below for detailed comments).

I recommend acceptance of the paper because of its strengths listed above, although I think the writing could be much improved to target a broader audience.

Main concerns:
1) Too much prior knowledge is assumed from the reader. I have worked at the intersection of theoretical neuro and ML for 8 years and am familiar with some recent literature on grid cell modeling, but still couldn't make sense of many aspects of the paper because of the many assumptions of prior knowledge. One exception is a fairly good description of the SR. But what is the  "Woodbury inversion formulae"? Shouldn't a barrier in the map break the translation invariance property assumed by the authors? What does CAN stand for? What does OI stand for?  What did we learn from comparing these models with the present model?
2) It is unclear how the "sense of direction" can be calculated in practice by a grid cell network. Do the authors assume a successive exploration of all possible directions, followed by an estimation of the corresponding probability of reaching the target? What would be the mechanistic implementation of this algorithm in a grid cell network? It would be helpful to see a concrete description or simulation of a neural network implementing action-directed SR.

Typos:
fig1 caption: "of of"
singel-neuron
framrework
discrinminative

---

### Official Review · AnonReviewer4 · 2020-10-28
**a simple model using grid cells for navigation**

**Rating:** 5
**Confidence:** 4

**Review:**

This paper proposed a model of navigation based on grid cells and the successor representation (SR).

Quality: This submission contains some interesting ideas. However, it feels that several key statements in the paper need to be clarified, and some statements need to be toned down. Overall, I feel this is a good submission, but the quality did not quite reach the bar in its current form.
Clarity: The writing requires some improvement. The key idea is not very clear. What does the “sense of direction” mean exactly? Also, it is mentioned that the model is capable of predicting the effect of arbitrary directed transitions, irrespective of local details, e.g., obstacles. This seems to suggest the grid cell responses are not affected by the obstacles. It that’s indeed the case, it would be inconsistent with experimental findings. (Am I misinterpreting this?)  Also, it is not clear exactly what are the inputs and outputs of the grid cell system.
Originality: Some key ideas presented in the paper (e.g, eigendecomposition of the transition matrix, successor representation) is fairly standard in previous literature, although the authors show that there is a way to unify some of the previous models with these notions.
Significance:  It is difficult to judge how much this work really adds to the previous literature.

Pro:
- the paper shows that a simple model based on grid cells and SR representation can perform navigation in some simple environments.
- the paper attempts to unify various kinds of grid cell models, which is interesting.
- the model is mathematically quite elegant and simple.

Cons:
- Some key ideas in the paper, e.g. eigen-decomposition of transition matrix, SR representation, are not new.
- The “sense of direction” added to the model is not clearly described. It is unclear how it is implemented in the network model, and how this information can get to the grid cells.
- It would be useful if the novel contributions could be more clearly stated. For example, Section  2 seems to be pretty standard materials. The paper would benefit by highlighting the true innovations (assuming there are some).
-Furthermore, the claim of unifying previous models is an over-statement and potentially misleading. Clearly, the proposed model is related to many of the previous models, but to go one step further and say that it unifies these previous models, that would seem to be a over-claim in my view.  For example, the proposed model does not have any recurrent connections- it is difficult to see how this unify the CAN models which crucially study  the role of recurrent computations.

Concerns:
1) It is claimed that the proposed model generalizes across  different environments. Does that predict that the grid cells are shared across the different environment? Do the local cues such as barriers and reward locations influence the firing pattern of grid cells? Are the predictions of the model consistent with neurophysiological data?
2) A key quantity in generating “a sense of direction” is the quantity s_G in Eq. (9). However, I didn’t find anything about how it is defined. Sorry if I am missing something. How is S_G encoded in the network? How would the grid cells have access to this information?
3) The paper claims that this model also unifies the OI models. Notably, various experimental results have argued against the OI models, which now lacks experimental support. Thus, it is unclear how much this really adds.
4) How does the model update the SR information when environment changes (e.g., inserting a barrier)?
5) It is stated that  “a computational role for the neural grid codes: generating a "sense of direction" (eq. 9) even in new or bounded environments, via utilising a Fourier basis for a larger toroidal pseudo space”. What specific predictions does this model make for neurophysiology? Is there a way to falsify the hypothesis/model?

A few more clarification questions:

How does Fig F shows “SR values can be used for gradient-based navigation”?

How is the barrier implemented using “wind” exactly?

What does “intuitive planning” mean?

I am puzzled how theta phase precession could arise from the model. Apologize if this is obvious…Could the authors flesh out the ideas a bit more?

---

### Official Review · AnonReviewer3 · 2020-10-28
**Prediction & Generalization Grid Cells**

**Rating:** 7
**Confidence:** 4

**Review:**

Summary: The authors propose as extension of the successor-representation approach to Grid cells. The paper shows that this model can generate several experimentally observed properties of grid cells, and can be used in navigation of novel/mutable environments. Overall, the work should be of interest to any ICLR attendees who engage in research surrounding grid cells.

Strong Points:
Overall, the paper is very well organized and written. The presentation of the proposed model being able to unite previous normative and mechanistic models is elegant.
Extreme care is taken to show that the model generates experimental findings such as phase precession.

Weak Points:
In section 4.2, the authors relate the fourier modes from the previous sections. Given that in the previous section, the authors propose grid cells as being weighted Fourier modes (eq 10), I worry that there might be explicit band cells  in the proposed model. Given that these neurons have been strongly questioned in the electrophysiology literature, this could be a huge impediment to the model, as with previous implementations of OIM. If instead, the fourier component is meant to be distributed among the input population, it the authors should make this clear

---

### Official Review · AnonReviewer5 · 2020-11-10
**Interesting but too limited**

**Rating:** 4
**Confidence:** 4

**Review:**

This paper extends the intuitive planning methodology with Fourier analysis to predict state occupancy of directed transitions on a graph-structured state space. The paper uses this idea to develop new connections with path-integration grid cell models for their prediction, illustrating the method in a gridworld and simulated grid cell domain.

I am leaning to reject this paper. On the one hand I agree with the authors on the importance. On the other hand, the contribution feels limited, as it seems to apply almost exclusively to 2D gridworlds and velocity-based actions.  Additionally, many of the insights here about Fourier representations are simple, and the useful connections to reinforcement learning are well-known from the proto-value function work (Mahadevan, 2005). The current paper could use some organizational improvements, and the experiments could be more comprehensive and tuned to demonstrate sharper points.  All taken together, the paper needs a bit more work to provide a significant contribution to the machine learning community. I believe another round of editing, tightening the material, and improving the results could indeed make this a very solid paper.

Current justification:
The proposed method seems to have limited practical value. The demonstrations in Figures 1, 2, and 3 are made in small gridworlds -- 50x50 is the largest. The paper mentions how results are very sensitive to stochasticity; so it can be assumed it applies only to deterministic gridworlds. Section 3 suggests the transition structure can only derive from a 2D grid space, and that the method requires a periodic boundary condition hold. When it doesn’t, the suggested fix is to inflate the state space to at least twice its size. This feels quite restrictive and potential wasteful in terms of computation. In summary, the method only works on deterministic gridworlds where certain boundary conditions hold or the space is small enough to double its size so it still fits in memory. Can the authors provide examples of more interesting domains where this strategy is feasible? Can they demonstrate that there are sufficiently rich problems in the gridworld space where this method provides significant utility? If there are examples from biology, then explain what those are and show that the Fourier approach provides a significant enhancement in some regard.

Currently, a reader would not be able to replicate the presented results. Figure 1 shows transition matrices and the histograms of two row slices. The domain details needed to generate these images were not explained. It would be helpful to know: 1. What state space defines the problem, 2. How 1.E was generated, and additionally 3. What the colors represent. The same general comment applies to the other figures. Searching through the appendix did not return any extra information. How were the grid cells simulated? Even if you do not release code, the paper should still contain enough information to replicate the results.

The presented data does not test any hypotheses. The first two figures are marked as demonstrative. The last two figures mostly show qualitative information from the successor representation and illustrations of gridworlds. There are no baselines with which to make comparisons. Figure 3.H provides the only piece of quantitative data, and it implies the prediction error increases with transition noise? Can the authors elaborate on what point is being made in Figure 3.H? How was the data gathered and why does it appear to increase monotonically in discrete steps?

The writing also needs to be tightened in several places. In the discussion, the paper claims “the resulting prediction framework is computationally efficient.” However, the paper provides no evidence for this claim. And along these lines, the data that is presented seems disconnected from any logical argument in the paper -- at least one that I could track. A simple editing pass will reveal many spelling errors and typos, e.g. “theorem to calculaet.”

Overall the paper is too disorganized, unfocused, and limited to merit acceptance. I look forward to the author response so that I can refine both my understanding and assessment of this work.

---

### Decision · Program_Chairs · 2021-01-07
**Final Decision**

**Decision:**

Accept (Poster)

**Comment:**

This paper was controversial amongst the reviewers. There is clear utility to the ICLR community: a new model of grid cells based on well-known technique (SR) used frequently in ML; good science---careful analysis showing the proposed model exhibits key properties and useful in synthetic navigation domains;  such work reminds of the important concerns in natural learning systems which is relevant to those that wish to simulate and build intelligence. Two of the reviewers with subject matter experience in the area advocated for acceptance.

On the other hand, many readers of ICLR may find the paper confusing and unsatisfying as some of the reviewers did. The empirical work was limited to small domains and mostly in the form of demonstrations---a typically ICLR reader would expect a performance improvement claim or a scientific hypothesis tested by each experiment. Presented as a new algorithm for ML the paper might appear too limited and simple (e.g., relying on state aggregation). The reviewers with neuro background found the paper clear and well organized, while the ML reviewers found it confusing. The relevance of the work will be limited to a smaller subset of researchers---but this is true of many ML works also. Finally, ML readers might be more familar with neuro work which propose computational models and then validate those models against real neural activity data from brains. This is work is not like that, rather using synthetic data to demonstrate important properties and explore empirical conjectures about the model.

In the end the paper is boarder line: the subject matter experts both listed issues that should be addressed (e.g., band cells issue), while the reaction of the ML reviewers suggests the impact of the work might be reduced at ICLR (compared to other venues). Additional text clearly articulating the scope and managing reader expectation could mitigate this concern, but it's not a small task to change the tone and pitch this way. Scientific conferences are about insights and understanding, this paper provides both. Please consider the suggested edits to maximize the impact of your work at ICLR this year.